# Real-World Evidence—Current Developments and Perspectives

**DOI:** 10.3390/ijerph191610159

**Published:** 2022-08-16

**Authors:** Friedemann Schad, Anja Thronicke

**Affiliations:** 1Interdisciplinary Oncology and Palliative Care, Hospital Gemeinschaftskrankenhaus Havelhöhe, 14089 Berlin, Germany; 2Research Institute Havelhöhe, Hospital Havelhöhe, 14089 Berlin, Germany

**Keywords:** review, real-world evidence, real-world data, randomized controlled trials, registry, digital health technology, early drug approval

## Abstract

Real-world evidence (RWE) is increasingly involved in the early benefit assessment of medicinal drugs. It is expected that RWE will help to speed up approval processes comparable to RWE developments in vaccine research during the COVID-19 pandemic. Definitions of RWE are diverse, marking the highly fluid status in this field. So far, RWE comprises information produced from data routinely collected on patient’s health status and/or delivery of health care from various sources other than traditional clinical trials. These sources can include electronic health records, claims, patient-generated data including in home-use settings, data from mobile devices, as well as patient, product, and disease registries. The aim of the present update was to review current RWE developments and guidelines, mainly in the U.S. and Europe over the last decade. RWE has already been included in various approval procedures of regulatory authorities, reflecting its actual acceptance and growing importance in evaluating and accelerating new therapies. However, since RWE research is still in a transition process, and since a number of gaps in this field have been explored, more guidance and a consented definition are necessary to increase the implementation of real-world data.

## 1. Introduction

Real-world evidence (RWE) research has received attention in the recent decade and is playing an increasing role in health care decisions [1]. As shown by the recent COVID-19 pandemic, RWE may help to broaden the view on patient populations and multiple medications not being included or implemented in classical clinical trials [2]. Therefore, the interest in whether non-randomized RWE can help to supplement the evidence of randomized controlled trials (RCTs) or aid in clinical decision-making is growing. Globally, growing activities of international initiatives and the release of legal frameworks by regulatory authorities have been observed in this field [3]. The British Medicines and Healthcare Products Regulatory Agency (MHRA) defines RWE as information produced from data routinely collected on patients, such as electronic health records (EHR) as well as disease and patient registries [3]. The Food and Drug Administration (FDA) defines RWE as “clinical evidence regarding the usage and potential benefits or risks of a medical product derived from analysis of RWD” [4]. The US Congress defines real-world data (RWD) utilized in RWE as “data regarding the usage, or the potential benefits or risks, of a drug derived from sources other than traditional clinical trials” [3]. The FDA has expanded the definition of RWD as follows: RWD “are data relating to patient health status and/or the delivery of health care routinely collected from a variety of sources” [3]. Thus, RWE research is based on a number of sources, including EHRs, claims and billing activities, laboratory data, hospital data, product and disease registries, patient-generated data including in home-use settings, data gathered from other sources that can inform on health status (e.g., mobile devices), and data linkage approaches [2,5]; see Figure 1. Information generated and analyzed from RWE can help in considering the effectiveness, efficacy, and safety of a medicinal product or an electronic device. Thus, the potential of high-quality RWD seems promising—for research as well as for health care. For decades, RWE has played a role in post-approval, but not in approval processes. 

The present review indicates, in its first chapter, current RWE gaps and highlights the chronological development of RWE, including the actions taken and guidelines released by the regulatory authorities. The second chapter describes the current situation of RWD in accelerating the orphan drug market and early drug assessment. In the third chapter, the contribution of real-world results during the COVID-19 pandemic and how it impacted the understanding of vaccine research is described; the fourth chapter reviews the importance of high-quality cancer registries as a growing RWE source in oncological research. The following two chapters, chapters five and six, explore the ways in which RWE facilitates RCTs and digital health technology, respectively. In chapter seven, four FDA-funded projects of RWE are introduced. The current review generates a chronological overview of RWE development over the past decade and contextualizes lessons learned from research in oncology and the COVID-19 pandemic with actual perspectives from clinicians.

## 2. Results & Discussion

### 2.1. Developments and Regulatory Guidelines of RWE

As RWE has received increased attention over the past decade [6,7,8], several RWE initiatives have been established, and regulatory authorities worldwide have disseminated guidance, framework, and standards in this field. In order to explore the potential of RWE, the FDA launched the 21st Century Cures Act [6] in 2016; see Table 1. 

The Cures Act had the intention to accelerate medicinal product development and to effectively delivering medicinal advances to patients. Subsequently, the FDA released a framework in 2018 to evaluate RWE sources [7,9].

In the following years, a series of several other guidelines for the use and submission of RWE documents were released, such as the guidance on the use of RWE to support regulatory decision-making for medical devices (2017) and the industry guidance on the use of electronic health record data in clinical investigations (2018) [10]; see Table 1. In 2019, the FDA published guidance on how to use evidence drawn from RWE to fulfill requirements for post-approval studies of drugs and biologics [12]; see Table 1. 

Meeting the gap of limited access to RWE due to multiple sources and interoperability, the European Medicines Agency (EMA) released a draft guidance on studies based on registry data as one source of RWD for marketing authorization applicants in 2020 [8]; see Table 1 and Figure 2. Recently, the absence of an appropriate RWE information submission structure was identified by the EMA as an additional gap: by reviewing applications for marketing authorization in 2018 and 2019, the EMA found that RWD was utilized in 40% of these applications, mainly at the post-approval stage and in 18% of extensions of indication filings [21]. Furthermore, it was observed that disease registries and hospital data were the two most frequently used data sources in applications for marketing authorization and indication expansions [21]. Several years ago, the EMA started to support and to encourage companies utilizing these disease registries as a source of RWE data [18]; see Table 1. The EMA plans to further analyzing the criteria used to evaluate RWE. In addition, further gaps have to be identified, e.g., whether a new RWE terminology is needed and whether RWE includes safety studies besides effectiveness studies [18]; see Figure 2. Further gaps of RWE research include the transparency of RWE studies. In accordance with this, the RWE Transparency Initiative [7] released a report on the improvement of transparency of RWE studies by a routine pre-registration of RWE study protocols following suggested guidelines in 2020 [22]; see Figure 2. In October 2021, the Real-World Evidence Registry was launched in co-operation with the Real-World Evidence Transparency Initiative, where RWE studies can be pre-registered [7,23]. Adding value to this topic, the International Society of Pharmacoepidemiology (ISPE), disseminating information on the effects of health interventions in humans, has recently formed a RWE taskforce [14] to provide guidance and develop protocols for study investigations and data extraction procedures in this field.

### 2.2. RWE Accelerates Approval in Early Benefit Assessment

It is believed that implementation of RWE in early assessments of newer therapies may speed up approval processes of effective therapies and reduce costs of drug development [24]. To present the additional benefit of RWE in the early benefit assessment of drugs, this topic has been included in the health policy arena in 2020 by the German joint federal committee, which commissioned the Institute for Quality and Efficiency in Health Care. Finally, a scientific rapid report on the evaluation of healthcare research for the purpose of the benefit assessment of drugs was released in 2020 [16,17]. The report concluded that conventional studies for rare diseases were often too short or did not collect patient-relevant endpoints. Therefore, the report suggested the inclusion of healthcare research data such as results from comparative studies, data collection of electronic patient records, as well as billing data from health insurance companies (all as per definition RWE sources) for the early benefit assessment of drugs. It has been recognized in this field that the data situation is insufficient for accelerated approval and for drugs implicated in the rare disease market [14]. Rare disease studies, being mostly limited to case studies or studies with small patient numbers, would greatly benefit from EHR and disease registries to gain insights into epidemiology, treatment, and outcomes in this field. In addition, rare diseases are currently under-represented in databases or records due to the older ICD-10 regulation. Given the new ICD-11 regulation in 2022, new and approximately 10-times more data on these diseases will become available, and it will also be possible to explore new treatment pathways due to RWE [25].

### 2.3. Contributions of RWE to COVID Research

During the COVID-19 pandemic, the number of scientific questions to be answered surpassed the capacity of the available means to conduct RCTs. RWE studies were the primary source of evidence during the COVID-19 pandemic that reported on patient symptoms and the influence of patient characteristics and risk of morbidity and mortality [2,26,27]. In addition, RWE contributed to the results supporting the wearing of the masks and other non-pharmacological interventions [28]. It was the COVID-19 pandemic that appeared to increasingly incorporate Bayesian models (which adaptively adjust the influence of external controls on the analysis of trial data) as statistical models to answer important questions such as the accuracy of diagnostic tests [29], the COVID-19 diagnosis itself [30], or the mortality of infected patients [2], among others. RWE studies have also accelerated vaccine efficacy and safety research in the field, as they relied on existing data that were more rapidly implemented. Hereby, data from classical post-licensing phase IV studies or more novel sources such as RWE registries or COVID-EHRs were used to monitor the long-term safety and the known or potential risk of vaccines [2]. The first real-world results on COVID-19 vaccine efficacy came from an Israelian study, stating a 94% real-world efficacy of the Pfizer/BioNTech vaccine for persons one week after their vaccination compared to non-vaccinated persons [31]. Other RWE studies followed, e.g., a study of the Center of Control Disease in the U.S. for people living in long-term care facilities given one dose of the Pfizer/BioNTech vaccine with a successful real-world vaccine efficacy of 63% against infection [32]. The ongoing RWE study HERO-Together being conducted by the Duke Clinical Research Institute, which is collecting data from healthcare workers and their families and community for up to two years after vaccination, evaluates long-term side effect rates [33]. Further long-term results on COVID-19 vaccine hesistancy, efficacy, combinations, and safety will be expected from high-quality RWE research. 

### 2.4. Contributions of RWE to Cancer Research

A first systematic review evaluated oncological new drug applications and biologics license application between 2015 and 2020 [34] and found that, among other results, already 11 among 133 oncology new therapy approvals utilized RWE to support the efficacy of the drugs. The review concluded “that real-world studies used as external controls complemented efficacy data from single-arm trials in successful oncology product approvals, and that the key attributes identified include early engagement, a priori protocol development, and robust research design” [34]. Given the help of RWE research, the utility and application of cancer therapies can be studied, e.g., in second- or third-line treatment where several equivalent options exist. A recent real-world analysis showed that nivolumab rather than cabozantinib was administered as a second line to metastatic renal cell carcinoma (mRCC) patients, even though there were no criteria of favoring one or the other on the basis of former RCT results, and real-world cost-effectiveness of both treatments seemed to be similar [35]. In another RWE study in mRCC patients, those with a clear cell histology and good risk features had a significantly longer progression-free survival with cabozantinib—thus, certain subgroups of patients may benefit from cabozantinib rather than from nivolumab [36]. In lung cancer, RWE adds to lacking information on safety and survival outcome of immunotherapy in elderly patients (>75 years) or patients with ECOG performance status 2, as shown by a recent review [37]. While RWE studies support the use of immunotherapy in elderly patients, the data cautions on the application of checkpoint inhibitors in performance status 2 patients due to a lacking survival benefit [37,38]. Such lacks being detected by RWE research in turn leads to actions by international cancer associations and regulatory bodies to call for RCTs matching real-world situations more closely [38]. Thus, it is expected in the future that RWE will also help to speed up approval processes, bringing new therapies to the patients, especially in the orphan and oncological drug development. In accordance to RWE’s role during the COVID-19 pandemic, the majority of American oncologists believe that RWE will improve efficiency of clinical trials for cancer drug development [39]. In line with this, cancer institutes and pharmacological companies increasingly include an external RWD control arm in their clinical trials in order to accelerate therapies being provided to patients. Among other issues, this may also help ameliorating the diversity problems of representative patients in oncological trials. An emerging field of RWE in oncology represent cancer registries. This may be partly because the primary data collection and analysis in non-interventional studies using registry data may be faced with less hurdles than using data from EHRs, medical claims, or other data sources [40]. Cancer registries indicate the degree of implementation of therapies and the use of guidelines in healthcare, and *vice versa,* feed data from healthcare into guideline updates [41]. Already more than 700 cancer registries exist worldwide exploring RWE cancer entities, population, standard-oncological treatments, and their associations [42]. In the near future, the guidance on data quality, accuracy, completeness, provenance, and traceability of registry data as primary data in non-interventional studies or arms of clinical studies will be key for RWE development [40]. Using high-quality patient registries, both with and without randomization, relevant results for disease assessments can be received, according to a report in 2020 of the German joint federal committee, the highest decision-making body of the German health care system [17].

### 2.5. RWE and RCTs—One Aim

In 2020, the MHRA released guidance on RCTs generating RWE to support regulator decisions [20]; see Table 1. Together with further upcoming guidance documents in this field it aimed to explore the potential of RWE in the support of regulatory submission, thereby covering topics as authorization, design, and database quality. Importantly, the MHRA concludes that RWE generated from RCTs including RWD “is not generally considered of more or less value for regulatory decision making than evidence from conventional RCTs provided if the data quality is robust and the trial well designed” [43]. The perception pattern of RWE has shifted towards being the “most important data for informing treatment decisions, followed by clinical trial data” as a result of a grand survey of 866 US physicians from 50 states [39]. Growing initiatives of including RWE results in clinical and therapeutic guidelines have been shown, e.g., in the UK, where 43 different RWE studies on 12 disease fields based on EHR data were utilized for the development of the NICE clinical practice guidelines [44]. The authors conclude that “there seems to be an increasing trend in the use of healthcare system data to inform clinical practice, especially as the real world validity of clinical trials is being questioned” [44]. In addition, another review found that an increasing number of primary care databases, the key source of RWE in the UK, are used for NICE health technology assessments, to support current treatment, and for the determination of economic model entry parameters [45]. Treatment decisions on the basis of RWE in, e.g., pragmatic trials can help to consider certain patient sub-populations that benefit most effectively from a certain treatment which in turn leads to relevant cost savings [46,47]. 

It is widely acknowledged that RWE more likely reflects real treatment effects and is more generalizable than evidence from standardized clinical trial data [20]; see Figure 3. The strengths of RWE lie in mapping the reality of health care, providing insights into therapy course or particular subgroups with various comorbidities. In addition, it gives the opportunity to study diseases in very small or very special patient populations. Furthermore, the reduction of purely selected patient collectives and thus the effectiveness of interventions under routine conditions can be analyzed in RWE; see Figure 3. In addition, RWE studies help to provide information about disease risks, improve diagnostics, make over-use and under-use visible, and can be used in the development of experimental therapies [24]. Thus, RWE research is complementing RCTs for identifying research gaps and for the evaluation of treatment effectiveness, especially in diverse patient sub-groups, see Figure 3.

### 2.6. Digital Health Technologies 

Newer RWE methods including digital health technologies (DHTs), data linkages, and registries have an expanding role in health care, offering considerable opportunities for drug development [29]. DHTs comprise of wearable devices, mobile apps, software, computing platforms, sensors for healthcare, telehealth, telemedicine, and personalized medicine [48]. In Europe, currently (as of June 2020), a certain definition for DHTs has not been given by the EMA in order to not exclude any innovative approaches, reflecting the high dynamics in this field. However, the EMA has limited the requirements for DHTs to the “specific use of a methodology in the development, use or monitoring of medicinal products pre- or post/authorisation, evaluation and ultimately use of medicines” [15], such as digital record systems (e.g., apps), sensors, wearables, tele-healthcare, and health data analytics. DHTs may help to improve the prevention and early diagnosis of life-threatening diseases, as well as the management of chronic conditions outside of traditional health care settings and thus help to reduce costs and inefficiencies, improve access, and make medicine more personalized [48]. In 2018, an evidence standards framework for DHTs had been released by the British National Institute for Health and Care Excellence (NICE) to inform innovators and commissioners on the requirements for DHTs for the health care system and to create the approach and commissioning to be more dynamically focused on real values for patients [11]. In Europe, DHT applicants are broadly supported, and the EMA has released a Q&A document in 2020 to give guidance to the applicants during submission on whether to seek scientific or other regulatory advice [15].

### 2.7. FDA Funded RWE Research

Alongside the FDA framework to explore RWE sources, in November 2020, the FDA announced four grant awards on the utility of RWD to accelerate regulatory decision-making [19]; see Table 2. One project, as a co-operation of the Brigham and Women’s Hospital and Harvard Medical School, aims to show how RCTs and RWD have been linked and how the follow-up of patients can be extended beyond trial completion, thus marking new low-cost and highly effective approaches [49]. The second project, as a collaboration between the University of North Carolina and the company Genentech, aims at combining studies from different sources such as simulation studies, RCTs, and RWD studies and by providing training material, workshops, and R-alghorithms to be utilized for hybrid studies [50]. The third funded project as a collaboration of the Critical Path Institute and the Tufts Medical Center and the International Neonatal Consortium will collect data from neonatal intensive care units worldwide and deposit it in an RWD and Analytics Platform (RW-DAP) as they discovered a lack of a systematic collection and analysis of such data. Overcoming this lack would improve clinical trials in neonates [51]. Finally, a project of the company Verantos compares between various RWE methods in a RWD study to explore which of the methods are more valid and which yield credible results, and thus help the implementation of RWE within regulatory decisions [5].

## 3. Conclusions

The present review provides an overview of the different definitions, types, and utilities of RWE, how it has been developed over the last last decade, gaps and challenges, and the necessary regulatory guidances in this field to accelerate regulatory procedures and decision-making. In addition, it describes hurdles and potentials of RWE research, and contextualizes lessons learned from oncological research and the COVID-19 pandemic with the actual perception of clinicians. Hereby, the FDA in the US as well as the EMA in Europe play a pro-active role in exploring gaps and potentials of RWE and in guiding stakeholders in this field with the release of actual and appropriate guidelines. In addition, international RWE initiatives and registries have increasingly been developed supporting this process. Provided the data quality is robust and transparent, RWE research is complementing RCTs for identifying research gaps and for the evaluation of treatment effectiveness, especially in diverse patient sub-groups. Catalyzed by the COVID-19 pandemic, the general perception of RWE among clinicians is that it produces important data for informing treatment decisions. In addition, the ongoing implementation of RWE in early benefit assessments of newer therapies will potentially speed up approval processes of effective therapies and reduce costs of drug development. Newer research such as the implementation of registry data and various types of DHTs will grow in the near future, reflecting the dynamic state in this field. By funding RWE projects and by involving stakeholders in the development of future guidelines the trust, credibility, reproducibility, and transparency of RWE studies may further be promoted. This will help in speeding up and reducing costs of the delivery of patient-oriented effective therapies, adding beneficial value to public health.

## Figures and Tables

**Figure 1 ijerph-19-10159-f001:**
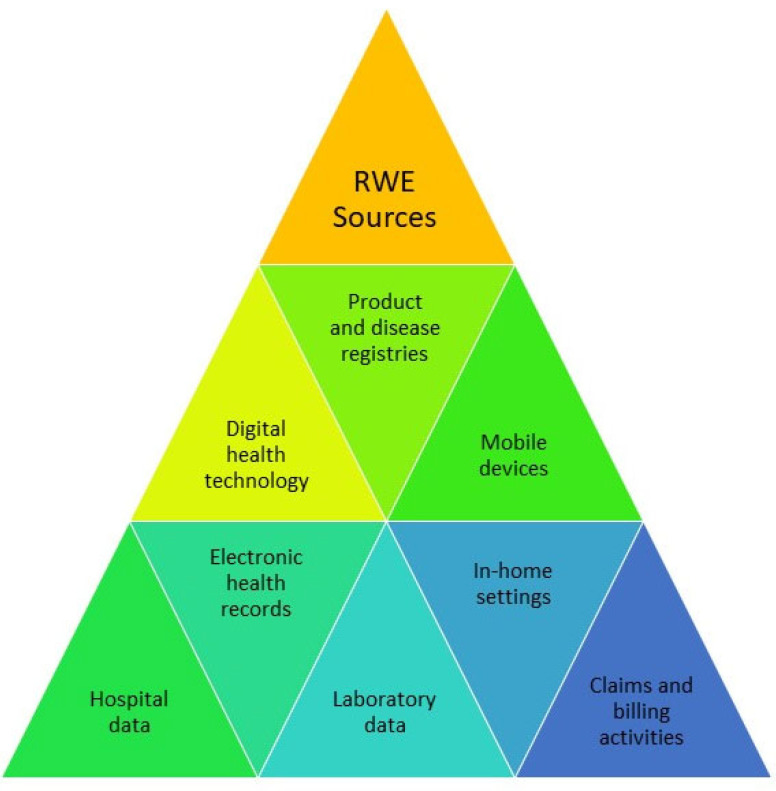
Sources of real-world evidence (RWE) research.

**Figure 2 ijerph-19-10159-f002:**
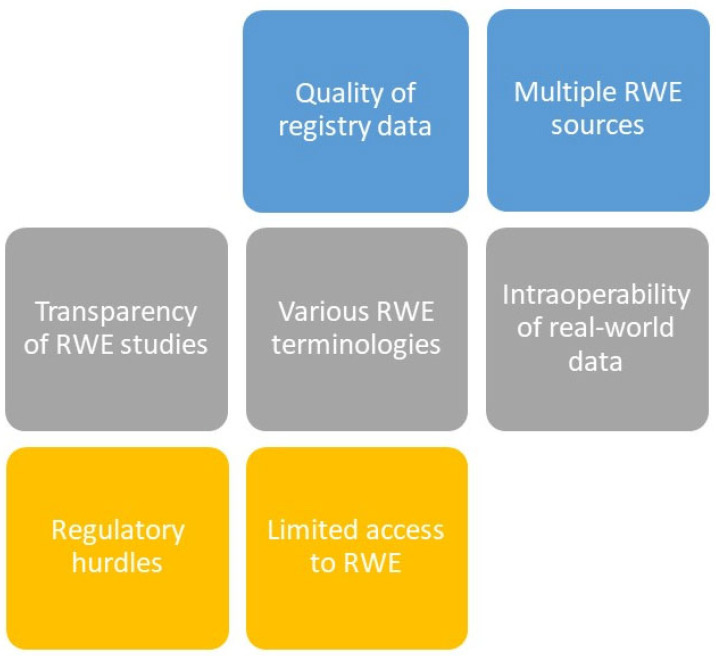
Challenges in RWE research.

**Figure 3 ijerph-19-10159-f003:**
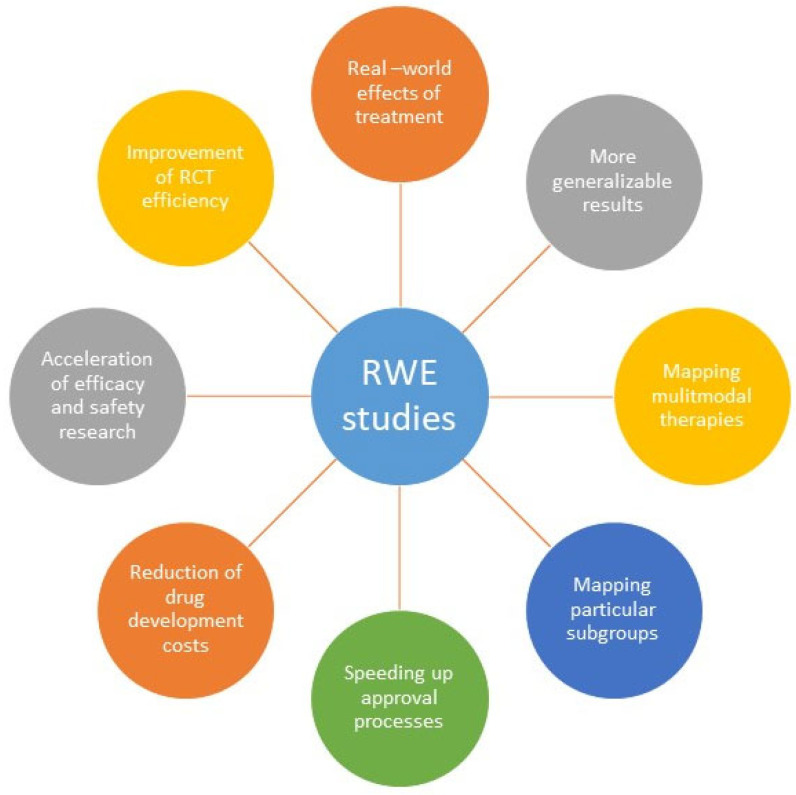
Potentials of RWE studies.

**Table 1 ijerph-19-10159-t001:** RWE developments in the international context for regulatory decision-making.

RWE Developments	Year
FDA—21st Century Cures Act [6]	2016
FDA—Use of RWE to Support Regulatory Decision-Making for Medical Devices [8]	2017
FDA—Use of Electronic Health Record Data in Clinical Investigations Guidance for Industry [10]	2018
FDA—Framework for FDA’s real-world evidence program [7]	2018
National Institute for Health and Care Excellence (NICE)—Evidence standards framework for digital health technologies [11]	2018
FDA—Submitting Documents Using RWD and RWE to FDA for Drugs and Biologics Guidance for Industry [12]	2018
HMA-EMA Joint Big Data Taskforce Phase II report: Evolving Data-Driven Regulation [13]	2019
ISPE’s Comments on the Core Recommendations in the Summary of the Heads of Medicines Agencies (HMA)—EMA Joint Big Data Task Force [14]	2019
EMA – Questions and answers: Qualification of digital technology-based methodologies to support approval of medicinal products [15]	2019
Federal Joint Committee (G-BA) (Germany) and the German Institute for Quality and Efficiency in Health Care (IQWIG): Scientific rapid report on the evaluation of healthcare research for the purpose of the benefit assessment of drugs [16,17]	2019
RWE taskforsk of the ISPE—Statement on Real-World Evidence (RWE) [7]	2020
EMA—guidance on registry-based studies as a tool to generate RWE for marketing authorization applicants and holder [18]	2020
FDA—grants for 4 projects exploring the utility of RWE [19]	2020
Medicines and healthcare products regulatory agency—Guidance on randomised controlled trials generating real-world evidence to support regulatory decisions [20]	2020
RAPS Euro Convergence—conference created by European regulatory professionals for regulatory professionals operating in Europe and other countries [21]	2021
FDA—Considerations for the Use of RWD and RWE to Support Regulatory Decision-Making for Drug and Biological Products, guidance for industry, draft guidance [4]	2021

**Table 2 ijerph-19-10159-t002:** FDA-funded RWE research projects for the utility of RWE.

RWE Developments	Institution
Enhancing evidence generation by linking randomized clinical trials (RCTs) to real-world data (RWD) [49]	Brigham and Women’s Hospital, Harvard Medical School, U.S.
Applying novel statistical approaches to develop a decision framework for hybrid randomized controlled trial designs which combine internal control arms with patients’ data from real-world data source [50]	University of North Carolina, Genentech, Inc., U.S.
Advancing standards and methodologies to generate real-world evidence from real-world data through a neonatal pilot [51]	Critical Path Institute (C-Path), Tufts Medical Center, International Neonatal Consortium, U.S.
Transforming real-world evidence with unstructured and structured data to advance tailored therapy (TRUST) [5]	Verantos, U.S.

## Data Availability

All relevant data are within the manuscript.

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
