# Peer review of "Real-World Evidence—Current Developments and Perspectives"

_ijerph, 2022, doi:10.3390/ijerph191610159_

Round 1

Reviewer 1 Report

I have deeply reviewed the manuscript. It is well designed and organized. Hence, the manuscript can be accepted in its present form.

In the paragraph, 2.3 :Cancer registries - RWE in oncology, you have mentioned that RWE has  accelerated the vaccine efficay, could you please explain more in detail the contribution of RWE in COVID research?

In the Conclusions as well, you have mentioned that RWE has provided more perception among clinicals in the COVID-19 circumstances, and it has  produced important data for informing treatment decisions. Could you please give examples of those? Is there any current vaccine or antiviral or drug available in the market for which RWE has contributed significantly?

Author Response

Response to Reviewer 1 Comments

Point 1: I have deeply reviewed the manuscript. It is well designed and organized. Hence, the manuscript can be accepted in its present form. In the paragraph, 2.3 :Cancer registries - RWE in oncology, you have mentioned that RWE has  accelerated the vaccine efficay, could you please explain more in detail the contribution of RWE in COVID research?
Response 1: Thank you for your review of our manuscript and your evaluation that it is ,well designed and organized,. According to your suggestion, we have explained more in detail the contribution of RWE in COVID research and have included a separate chapter 2.3 , Contributions of RWE in the COVID-19 research’ on page 4.

Point 2: In the Conclusions as well, you have mentioned that RWE has provided more perception among clinicals in the COVID-19 circumstances, and it has produced important data for informing treatment decisions. Could you please give examples of those? Is there any current vaccine or antiviral or drug available in the market for which RWE has contributed significantly?
Response 2: Thank you for this important point. We have now included further examples of how RWE has produced data for informing treatment decisions on page 7, line 243-254. As to your second question, there are current vaccines on the market, such as the Pfizer/BioNTech vaccine for which RWE has contributed significantly. Those data were important to overcome the vaccination hesitancy in the public. We have included this information in the new chapter 2.3 on page 5 and 6, lines 172-182.

Reviewer 2 Report

1. The content of the introduction is limited and does not give many details. The authors are suggested to add a few more details like the fundamental necessity of RWE. The research carried out on RWE and the importance of the current work.

2. Add a brief paragraph describing the article's flow at the end of chapter 1.

3. The authors stated that the interest in REW has been increasing over the decade; however, they have not given proper citations. Please include them.

4. It is hard to follow the review, so the authors are suggested to add more visual illustrations or tables for quick understanding.

Author Response

Response to Reviewer 2 Comments

Point 1: The content of the introduction is limited and does not give many details. The authors are suggested to add a few more details like the fundamental necessity of RWE. The research carried out on RWE and the importance of the current work.

Response 1: Thank you for the review of our manuscript. According to your suggestion, we have added more details on the fundamental necessity of RWE (page 1, line 27-31) and the importance of the current work (page 2 - introduction, line 60-63, as well as on page 9 - conclusion, line 314-317). As to the research carried out on RWE we have broaden the following paragraph (page 1, lines 42-46): “…Thus, RWE research is based on a  number of sources including EHRs, claims and billing activities, laboratory data, hospital data, product and disease registries, patient-generated data including in home-use settings, data gathered from other sources that can inform on health status (e.g. mobile devices), and data linkage approaches [3, 40]….” with an additional reference 40. We hope you agree.

Point 2: Add a brief paragraph describing the article's flow at the end of chapter 1.

Response 2: We have added a brief paragraph, according to your suggestions, at the end of the chapter 1, see page 2, lines 51-60.

Point 3: The authors stated that the interest in REW has been increasing over the decade; however, they have not given proper citations. Please include them.

Response 3: We have included a citation underlining our statement, see page 2, line 73 . As the FDA is giving strong signals about its interest in real-world research, stimulated in large part, by the 21st Century Cures Act we have included this act as one of three citations (reference 6) being supported by two other citations – The Use of Real-World Evidence to Support Regulatory Decision-Making for Medical Devices (FDA) as reference 36 and the Framework for FDA's real-world evidence program as reference 7.

.

Point 4: It is hard to follow the review, so the authors are suggested to add more visual illustrations or tables for quick understanding.

Response 4: Thank you. We have now included three visual illustrations for a quick and better understanding – one describing the RWE ressources (figure 1, page 2), the second the challenges in RWE (figure 2, page 5), the third the potentials of RWE (figure 3, page 7). Furthemore, we have included a table on the current FDA-funded RWE research projects (table 2, page 8).

Round 2

Reviewer 2 Report

Now, the authors have incorporated the changes pointed out by the reviewers, and the manuscript can be accepted for publication.